# Learning Hierarchical Semantic Image Manipulation through Structured Representations

**Seunghoon Hong**[†]     **Xinchen Yan**[†]     **Thomas Huang**[†]     **Honglak Lee**[‡,†]

[†]University of Michigan
[‡]Google Brain
[†]{hongseu,xcyan,thomaseh,honglak}@umich.edu    [‡]honglak@google.com

## Abstract

Understanding, reasoning, and manipulating semantic concepts of images have been a fundamental research problem for decades. Previous work mainly focused on direct manipulation on natural image manifold through color strokes, key-points, textures, and holes-to-fill. In this work, we present a novel hierarchical framework for semantic image manipulation. Key to our hierarchical framework is that we employ structured semantic layout as our intermediate representation for manipulation. Initialized with coarse-level bounding boxes, our structure generator first creates pixel-wise semantic layout capturing the object shape, object-object interactions, and object-scene relations. Then our image generator fills in the pixel-level textures guided by the semantic layout. Such framework allows a user to manipulate images at object-level by adding, removing, and moving one bounding box at a time. Experimental evaluations demonstrate the advantages of the hierarchical manipulation framework over existing image generation and context hole-filing models, both qualitatively and quantitatively. Benefits of the hierarchical framework are further demonstrated in applications such as semantic object manipulation, interactive image editing, and data-driven image manipulation.

## 1  Introduction

Learning to perceive, reason and manipulate images has been one of the core research problems in computer vision, machine learning and graphics for decades [1, 7, 8, 9]. Recently the problem has been actively studied in interactive image editing using deep neural networks, where the goal is to manipulate an image according to the various types of user-controls, such as color strokes [20, 32], key-points [19, 32], textures [26], and holes-to-fill (in-painting) [17]. While these interactive image editing approaches have made good advances in synthesizing high-quality manipulation results, they are limited to direct manipulation on natural image manifold.

The main focus of this paper is to achieve semantic-level manipulation of images. Instead of manipulating images on natural image manifold, we consider semantic label map as an interface for manipulation. By editing the label map, users are able to specify the desired images at semantic-level, such as the location, object class, and object shape. Recently, approaches based on image-to-image translation [3, 11, 24] have demonstrated promising results on semantic image manipulation. However, the existing works mostly focused on learning a *style* transformation function from label maps to pixels, while manipulation of *structure* of the labels remains fully responsible to users. The requirement on direct control over pixel-wise labels makes the manipulation task still challenging since it requires a precise and considerable amount of user inputs to specify the structure of the objects and scene. Although the problem can be partly addressed by template-based manipulation interface (*e.g.* adding the objects from the pre-defined sets of template masks [24], blind pasting of the object mask is problematic since the structure of the object should be determined adaptively depending on the surrounding context.

In this work, we tackle the task of semantic image manipulation as a hierarchical generative process. We start our image manipulation task from a coarse-level abstraction of the scene: a set of *semantic*

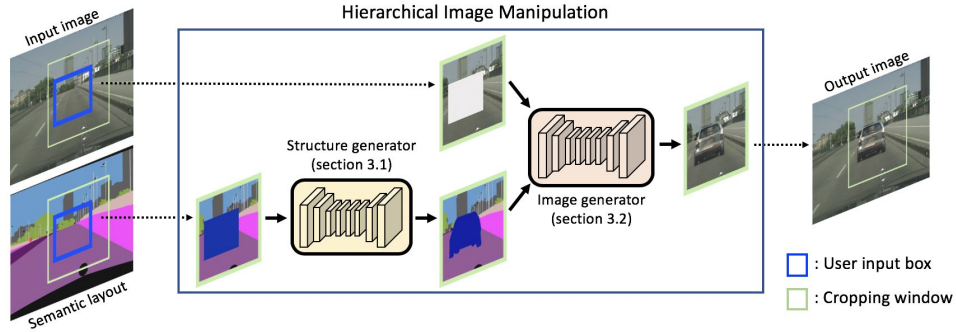

Figure 1: Overall pipeline of the proposed semantic manipulation framework.

bounding boxes which provide both semantic (what) and spatial (where) information of the scene in an object-level. Such representation is natural and flexible that enables users to manipulate the scene layout by adding, removing, and moving each bounding box. To facilitate the image manipulation from coarse-level semantic bounding boxes, we introduce a hierarchical generation model that predicts the image in multiple abstraction levels. Our model consists of two parts: layout and image generators. Specifically, our structure generator first infers the fine-grained semantic label maps from the coarse object bounding boxes, which produces structure (shape) of the manipulated object aligned with the context. Given the predicted layout, our image generator infers the style (color and textures) of the object considering the perceptual consistency to the surroundings. This way, when adding, removing, and moving semantic bounding boxes, our model can generate an appropriate image seamlessly integrated into the surrounding image.

We present two applications of the proposed method on interactive and data-driven image editing. In the experiment on interactive image editing, we show that our method allows users to easily manipulate images using object bounding boxes, while the structure and style of the generated objects are adaptively determined depending on the context and naturally blended into the scene. Also, we show that our simple object-level manipulation interface can be naturally extended to data-driven image manipulation, by automatically sampling the object boxes and generating the novel images.

The benefits of the hierarchical image manipulation are three-fold. First, it supports richer manipulation tasks such as adding, moving or removing objects through object-level control while the fine-grained object structures are inferred by the model. Second, when conditioned on coarse and fine-grained semantic representations, the proposed model produces better image manipulation results compared to models learned without structural control. Finally, we demonstrate the effectiveness of the proposed idea on interactive and automatic image manipulation.

## 2   Related Work

**Deep visual manipulation.**    Visual manipulation is a task of synthesizing the new image by manipulating parts of a reference image. Thanks to the emergence of generative adversarial networks (GANs) [6] and perceptual features discovered from deep convolutional neural networks [13, 21, 22], neural image manipulation [5, 14, 17, 20, 30, 32] has gained increasing popularity in recent years. Zhu et al. [32] presented an novel image editing framework with a direct constraint capturing real image statistics using deep convolutional generative adversarial networks [18]. Li et al. [14] and Yeh et al. [30] investigated semantic face image editing and completion using convolution encoder-decoder architecture, jointly trained with a pixel-wise reconstruction constraint, a perceptual (adversarial) constraint, and a semantic structure constraint. In addition, Pathak et al. [17] and Denton et al. [5] studied context-conditioned image generation that performs pixel-level hole-filling given the surrounding regions using deep neural networks, trained with adversarial and reconstruction loss. Contrary to the previous works that manipulate images based on low-level visual controls such as visual context [5, 14, 17, 30] or color strokes [20, 32], our model allows semantic control over manipulation process through labeled bounding box and the inferred semantic layout.

**Structure-conditional image generation.**    Starting from the pixel-wise semantic structure, recent breakthroughs approached the structure-conditional image generation through direct image-to-image translation [3, 11, 16, 33]. Isola et al. [11] employed a convolutional encoder-decoder network with conditional adversarial objective to learn label-to-pixel mapping. Later approaches improved the generation quality by incoorporating perceptual loss from a pre-trained classifier [3] or feature-

matching loss from multi-scale discriminators [24]. In particular, Wang et al. [24] demonstrated a high-resolution image synthesis results and its application to semantic manipulation by controlling the pixel-wise labels. Contrary to the previous works focusing on learning a direct mapping from label to image, our model learns hierarchical mapping from coarse bounding box to image by inferring intermediate label maps. Our work is closely related to Hong et al. [10], which generates an image from a text description through multiple levels of abstraction consisting of bounding boxes, semantic layouts, and finally pixels. Contrary to Hong et al. [10], however, our work focuses on manipulation of parts of an image, which requires incorporation of both semantic and visual context of surrounding regions in the hierarchical generation process in such a way that structure and style of the generated object are aligned with the other parts of an image.

## 3 Hierarchical Image Manipulation

Given an input image $I^{in} \in \mathbb{R}^{H \times W \times 3}$, our goal is to synthesize the new image $I^{out}$ by manipulating its underlying semantic structure. Let $M^{in} \in \mathbb{R}^{H \times W \times C}$ denotes a semantic label map of the image defined over $C$ categories, which is either given as ground-truth (in training time) or inferred by the pre-trained visual recognition model (in testing time). Then our goal is to synthesize the new image by manipulating $M^{in}$, which allows the semantically-guided manipulation of an image.

The key idea of this paper is to introduce an object bounding box $B$ as an abstracted interface to manipulate the semantic label map. Specifically, we define a controllable bounding box $B = \{\mathbf{b}, c\}$ as a combination of box corners $\mathbf{b} \in \mathbb{R}^4$ and a class label $c = \{0, \ldots, C\}$, which represents the location, size and category of an object. By adding the new box or modifying the parameters of existing boxes, a user can manipulate the image through adding, moving or deleting the objects[1]. Given an object-level specification by $B$, the image manipulation is then posed as a hierarchical generation task from a coarse bounding box to pixel-level predictions of structure and style.

Figure 1 illustrates the overall pipeline of the proposed algorithm. When the new bounding box $B$ is given, our algorithm first extracts the local observations of label map $M \in \mathbb{R}^{S \times S \times C}$ and image $I \in \mathbb{R}^{S \times S \times 3}$ by cropping squared windows of size $S \times S$ centered around $B$. Then conditioned on $M$, $I$ and $B$, our model generates the manipulated image by the following procedures:

- Given a bounding box $B$ and the semantic label map $M$, the structure generator predicts the manipulated semantic label map by $\hat{M} = G^M(M, B)$ (Section 3.1)
- Given the manipulated label map $\hat{M}$ and image $I$, the image generator predicts the manipulated image $\hat{I}$ by $\hat{I} = G^I(\hat{M}, I)$ (Section 3.2)

After generating the manipulated image patch $\hat{I}$, we place it back to the original image to finish the manipulation task. The manipulation of multiple objects is performed iteratively by applying the above procedures for each box. In the following, we explain the manipulation pipeline on a single box $B$.

### 3.1 Structure generator

The goal of the structure generator is to infer the latent structure of the region specified by $B = \{\mathbf{b}, c\}$ in the form of pixel-wise class labels $\hat{M} \in \mathbb{R}^{S \times S \times C}$. The outputs of the structure generator should reflect the class-specific structure of the object defined by $B$ as well as interactions of the generated object with the surrounding context (*e.g.* a person riding a motorcycle). To consider both conditions in the generation process, the structure generator incorporates the label map $M$ and the bounding box $B$ as inputs and performs a conditional generation by $\hat{M} = G^M(M, B)$.

Figure 2 illustrates the overall architecture of the structure generator. The structure generator takes in the masked layout $\bar{M} \in \mathbb{R}^{S \times S \times C}$ and the binary mask $\bar{B} \in \mathbb{R}^{S \times S \times 1}$, where $\bar{M}_{ijc} = 1$ and $\bar{B}_{ij} = 1$ for all pixels $(i, j)$ inside the box for class $c$. Given these inputs, the model predicts the manipulated outcome using two decoding pathways.

Our design principle is motivated by generative layered image modeling [19, 23, 27, 28, 29], which generates foreground (*i.e.* object) and background (*i.e.* context) using separate output streams. In our model, the *foreground* output stream produces the predictions on binary object mask $\hat{M}_{\text{obj}} \in \mathbb{R}^{S \times S \times 1}$, which defines the object shape tightly bounded by object box $B$. The *background* output stream produces per-pixel label maps $\hat{M}_{\text{ctx}} \in \mathbb{R}^{S \times S \times C}$ inside $B$.

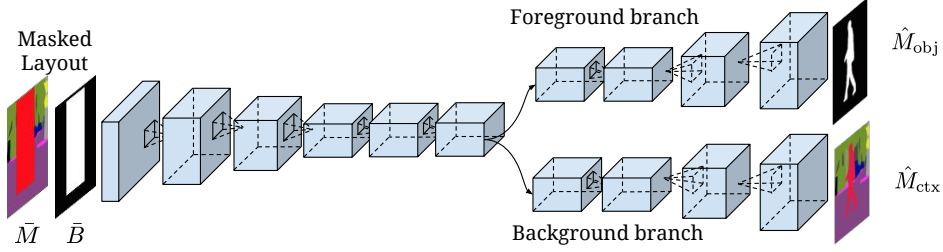

Figure 2: Architecture of the structure generator. Given a masked layout $\bar{M}$ and a binary mask $\bar{B}$ encoding the class and location of the object, respectively, the model produces the manipulated layout $\hat{M}$ by the outputs from the two-stream decoder corresponding to the binary mask of object and semantic label map of entire region inside the box.

The objective for our structure generator is then given by

$$\mathcal{L}_{\text{layout}} = \mathcal{L}_{\text{adv}}(\hat{M}_{\text{obj}}, M^*_{\text{obj}}) + \lambda_{\text{obj}}\mathcal{L}_{\text{recon}}(\hat{M}_{\text{obj}}, M^*_{\text{obj}}) + \lambda_{\text{ctx}}\mathcal{L}_{\text{recon}}(\hat{M}_{\text{ctx}}, \bar{M}), \tag{1}$$

where $M^*_{\text{obj}}$ is the ground-truth binary object mask on $B$ and $\mathcal{L}_{\text{recon}}(\cdot, \cdot)$ is the reconstruction loss using cross-entropy. $\mathcal{L}_{\text{adv}}(\hat{M}_{\text{obj}}, M^*_{\text{obj}})$ is the conditional adversarial loss defined on object mask ensuring the perceptual quality of predicted object shape, which is given by

$$\mathcal{L}_{\text{adv}}(\hat{M}_{\text{obj}}, M^*_{\text{obj}}) = \mathbb{E}_{M^*_{\text{obj}}}\left[\log(D^M(M^*_{\text{obj}}, \bar{M}))\right] + \mathbb{E}_{\hat{M}_{\text{obj}}}\left[1 - \log(D^M(\hat{M}_{\text{obj}}, \bar{M}))\right], \tag{2}$$

where $D^M(\cdot, \cdot)$ is a conditional discriminator.

During inference, we construct the manipulated layout $\hat{M}$ using $\hat{M}_{obj}$ and $\hat{M}_{ctx}$. Contrary to the prior works on layered image model, we selectively choose outputs from either foreground or background streams depending on the manipulation operation defined by $B$. The output $\hat{M}$ is given by

$$\hat{M} = \begin{cases} \hat{M}_{\text{ctx}} & \text{if } c = 0 \;\; (\textit{deletion}) \\ (\hat{M}_{\text{obj}}\mathbb{1}_c) + (\mathbf{1} - (\hat{M}_{\text{obj}}\mathbb{1}_c)) \odot M & \text{otherwise } (\textit{addition}) \end{cases}, \tag{3}$$

where $c$ is the class label of the bounding box $B$ and $\mathbb{1}_c \in \{0, 1\}^{1 \times C}$ is the one-hot encoded vector of the class $c$.

### 3.2 Image generator

Given an image $I$ and the manipulated layout $\hat{M}$ obtained by the structure generator, the image generator outputs a pixel-level prediction of the contents inside the regions defined by $B$. To make the prediction being semantically meaningful and perceptually plausible, the output from the image generator should reflect the semantic structure defined by the layout while being coherent in its style (*e.g.* color and texture) with the surrounding image. We design the conditional image generator $G^I$ such that $\hat{I} = G^I(I, \hat{M})$, where $I, \hat{I} \in \mathbb{R}^{S \times S \times 3}$ represent the local image before and after manipulation with respect to bounding box $B$.

Figure 3 illustrates the overall architecture of the image generator. The model takes the masked image $\bar{I}$ whose pixels inside the box are filled with 0 and the manipulated layout $\hat{M}$ as inputs, and produces the manipulated image $\hat{I}$ as output. We design the image generator $G^I$ to have a two-stream convolutional encoder and a single-stream convolutional decoder connected by an intermediate feature map $F$. As we see in the figure, the convolutional encoder has two separate downsampling streams, which we referred to as image encoder $f_{\text{image}}(\bar{I})$ and layout encoder $f_{\text{layout}}(\hat{M})$, respectively. The intermediate feature $F$ is obtained by an element-wise feature interaction layer gated by the binary mask $B_F$ on object location.

$$F = f_{\text{layout}}(\hat{M}) \odot B_F + f_{\text{image}}(\bar{I}) \odot (\mathbf{1} - B_F). \tag{4}$$

Finally, the convolutional decoder $g_{\text{image}}(\cdot)$ takes the fused feature $F$ as input and produces the manipulated image through $\hat{I} = g_{\text{image}}(F)$. Note that we use the ground-truth layout $M$ during model training but the predicted layout $\hat{M}$ is used at the inference time during testing.

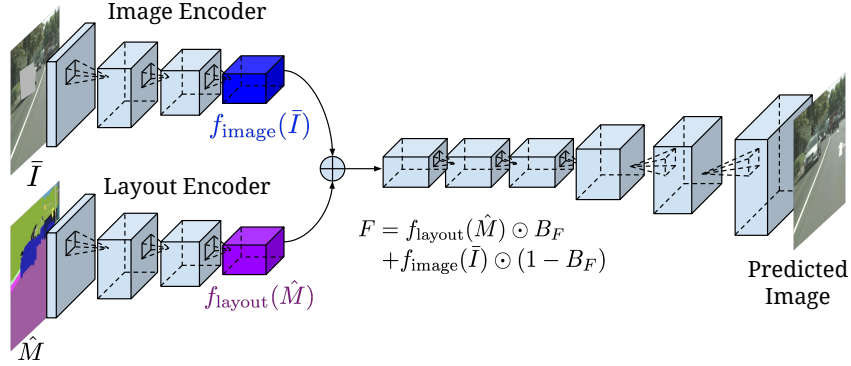

Figure 3: Architecture of the image generator. Given a masked image $\bar{I}$ and the semantic layout $\hat{M}$, the model encodes visual style and semantic structure of the object using separate encoding pathways and produces the manipulated image.

We define the following loss for the image generation task.

$$\mathcal{L}_{\text{image}} = \mathcal{L}_{\text{adv}}(\hat{I}, I) + \lambda_{\text{feature}}\mathcal{L}_{\text{feature}}(\hat{I}, I). \tag{5}$$

The first term $\mathcal{L}_{\text{adv}}(\hat{I}, I)$ is the conditional adversarial loss defined by

$$\mathcal{L}_{\text{adv}}(\hat{I}, I) = \mathbb{E}_I\left[\log(D^I(I, \hat{M}))\right] + \mathbb{E}_{\hat{I}}\left[1 - \log(D^I(\hat{I}, \hat{M}))\right]. \tag{6}$$

The second term $\mathcal{L}_{\text{feature}}(\hat{I}, I)$ is the feature matching loss [24]. Specifically, we compute the distance between the real and manipulated images using the intermediate features from the discriminator by

$$\mathcal{L}_{\text{feature}}(\hat{I}, I) = \mathbb{E}_{I, \hat{I}}\sum_{i=1}\|D^{(i)}(I, \hat{M}) - D^{(i)}(\hat{I}, \hat{M})\|_F^2, \tag{7}$$

where $D^{(i)}$ is the outputs of $i^{th}$ layer in discriminator, $\|\cdot\|_F$ is the Frobenius norm.

**Discussions.** The proposed model encodes both image and layout using two-stream encoding pathways. With only image encoder, it simply performs image in-painting [17], which attempts to fill the hole with patterns coherent with the surrounding region. On the other hand, our model with only layout encoder becomes similar to image-to-image translation models [3, 11, 24], which translates the pixel-wise semantic label maps to the RGB pixel values. Intuitively, by combining information from both encoders, the model learns to manipulate images that reflect the underlying image structure defined by the label map with appearance patterns naturally blending into the surrounding context, which is semantically meaningful and perceptually plausible.

## 4 Experiments

### 4.1 Implementation Details

**Datasets.** We conduct both quantitative and qualitative evaluations on the Cityscape dataset [4], a semantic understanding benchmark of European urban street scenes containing 5,000 high-resolution images with fine-grained annotations including instance-wise map and semantic map from 30 categories. Among 30 semantic categories, we treat 10 of them including `person`, `rider`, `car`, `truck`, `bus`, `caravan`, `trailer`, `train`, `motorcycle`, `bicycle` as our foreground object classes while leaving the rest as background classes for image editing purpose. For evaluation, we measure the generation performance on each of the foreground object bounding box in 500 validation images. To further demonstrate the image manipulation results on more complex scene, we also conduct qualitative experiments on bedroom images from ADE20K dataset [31]. Among 49 object categories frequently appearing in a bedroom, we select 31 movable ones as foreground objects for manipulation.

**Training.** As one challenge, collecting ground-truth examples before and after image manipulation is expensive and time-consuming. Instead, we simulate the addition ($c \in \{1, ..., C\}$) and deletion ($c = 0$) operations by sampling boxes from the object and random image regions, respectively. For training, we employ an Adam optimizer [12] with learning rate 0.0002, $\beta_1 = 0.5$, $\beta_2 = 0.999$ and linearly decrease the learning rate after the first 100-epochs for training. The hyper-parameters $\lambda_{\text{obj}}$, $\lambda_{\text{ctx}}$, $\lambda_{\text{feature}}$ are set to 10. Our PyTorch implementation will be open-sourced.

| | Layout | SSIM | Segmentation (%) | Human eval. (%) |
|---|---|---|---|---|
| SingleStream-Image | - | 0.285 | 59.6 | 12.3 |
| SingleStream-Layout | GT | 0.291 | 71.5 | 9.2 |
| SingleStream-Concat | GT | 0.331 | 76.7 | 23.2 |
| TwoStream | GT | **0.336** | **78.5** | **33.6** |
| TwoStream-Pred | Predicted | 0.299 | 77.9 | 20.7 |

Table 1: Comparisons between variants of the proposed method. The last two rows (TwoStream and TwoStream-Pred) correspond to our full model using ground-truth and predicted layout, respectively.

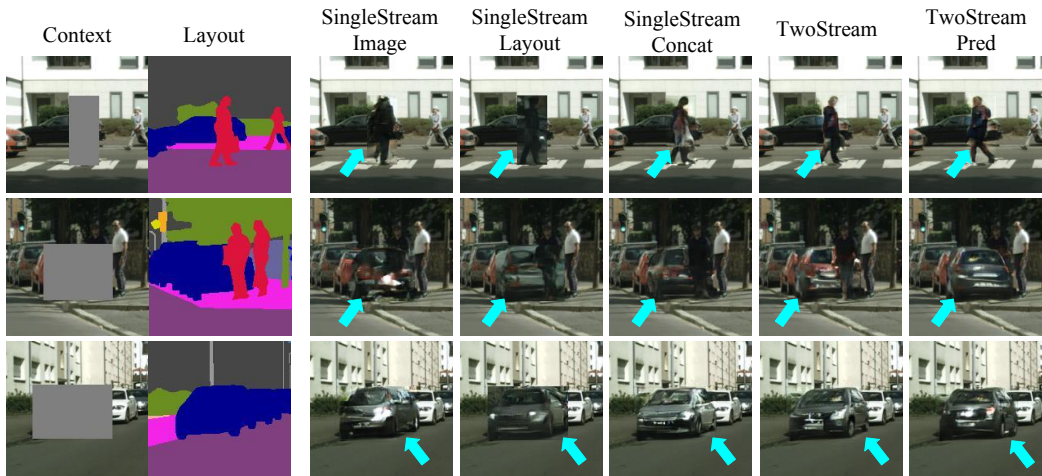

Figure 4: Qualitative comparisons to the baselines in Table 1. The first two columns show the masked image and ground-truth layout used as input to the models (except TwoStream-Pred). The manipulated objects are indicated by blue arrows. Best viewed in color.

**Evaluation metrics.** We employ three metrics to measure the perceptual and conditional generation quality. We use Structural Similarity Index (SSIM) [25] to evaluate the similarity of the ground-truth and predicted images based on low-level visual statistics. To measure the quality of layout-conditional image generation, we apply a pre-trained semantic segmentation model DeepLab v3 [2] to the generated images, and measure the consistency between the input layout and the predicted segmentation labels in terms of pixel-wise accuracy (layout → image → layout). Finally, we conduct user study using Mechanical Turk (AMT) to evaluate the perceptual generation quality. We present the manipulation results of different methods together with the input image and the class label of the bounding box and ask users to choose the best method based on how natural the manipulated images are. We collect the results for 1,000 examples, each of which is evaluated by 5 different Turkers. We report the performance of each method based on the ratio that method ranked as the best in AMT.

### 4.2 Quantitative Evaluation

**Ablation study.** To analyze the effectiveness of the proposed method, we conduct an ablation study on several variants of our method. We first consider three baselines of our image generator: conditioned only on image context (SingleStream-Image), conditioned only on semantic layout (SingleStream-Layout), or conditioned on both by concatenation but using a single encoding pathway (SingleStream-Concat). We also compare our full model using a ground-truth and the predicted layouts (TwoStream and TwoStream-Pred). Table 1 summarize the results.

As shown in Table 1, conditioning the generation with only image or layout leads to either poor class-conditional generation (SingleStream-Image) or less perceptually plausible results (SingleStream-Layout). It is because the former misses the semantic layout encoding critical information on which object to generate, while the later misses color and textures of the image that makes the generation results visually consistent with its surroundings. Combining both (SingleStream-Concat), the generation quality improves in all metrics, which shows complementary benefits of both conditions in image manipulation. In addition, a comparison between SingleStream-Concat and TwoStream shows that modeling the image and layout conditions using separate encoding pathways further improves the generation quality. Finally, replacing the layout condition from the ground-truth (TwoStream) to the predicted one (TwoStream-Pred) leads

| | Layout | SSIM | Segmentation (%) | Human eval. (%) |
|---|---|---|---|---|
| Context Encoder [17] | - | 0.265 | 26.4 | 1.1 |
| Context Encoder++ | GT | 0.269 | 35.7 | 1.4 |
| Pix2PixHD [24] | GT | 0.288 | **79.6** | 18.0 |
| TwoStream | GT | **0.336** | 78.5 | **79.5** |

Table 2: Quantitative comparison to other methods.

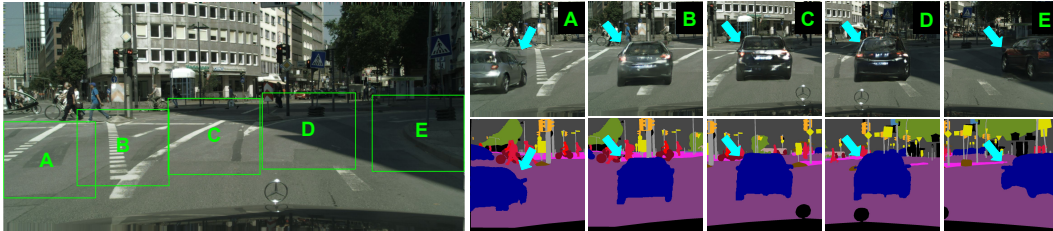

Figure 5: Generation results on various locations in an image.

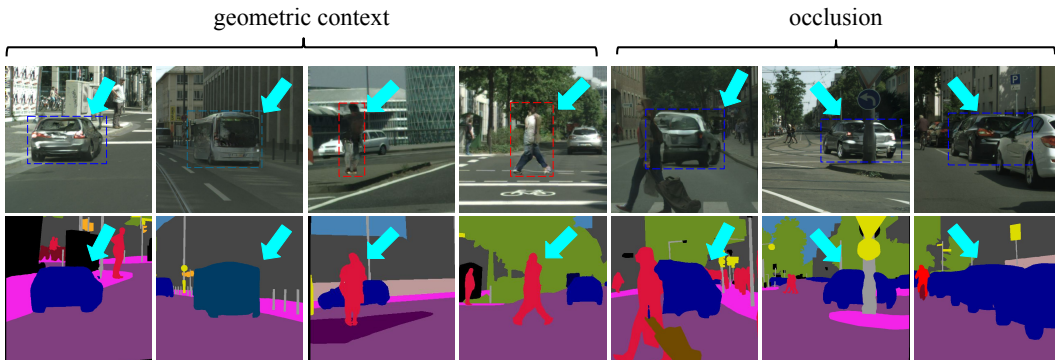

Figure 6: Generation results in diverse contexts.

to a small degree of degradation in perceptual quality partly due to the prediction errors in layout generation. However, clear improvement of `TwoStream-Pred` over `SingleStream-Image` shows the effectiveness of layout prediction in image generation.

Figure 4 shows the qualitative comparisons of the baselines. Among all variants, our `TwoStream` model tends to exhibit most recognizable and coherent appearance with the surrounding environment. Interestingly, our model with predicted layout `TwoStream-Pred` generates objects different from the ground-truth layout but still match the bounding box condition. such as a person walking in the different direction (the first row) and objects placed in different order (the second row).

**Comparison to other methods.** We also compare against a few existing work on context hole-filing and structure-conditioned image generation. First, we consider the recent work on high-resolution pixel-to-pixel translation [24] (referred as `Pix2PixHD` in Table 2). Compared to our `SingleStream-Layout` model, `Pix2PixHD` model generates an entire image from semantic layout. Second, we consider the work for context-driven image in-painting [17] (referred as `ContextEncoder`) as another baseline. Similar to our `SingleStream-Image`, `ContextEncoder` does not have access to the semantic layout during training and testing. For fair comparisons, we extended `ContextEncoder` so that it takes segmentation layout as additional input. We refer this model as `ContextEncoder++` in Table 2. As we see in the Table 2, our two-stream model still achieves the best SSIM and competitive segmentation pixel-wise accuracy. The segmentation accuracy of `Pix2PixHD` is higher than ours since it generates higher resolution images and object textures non-relevant to the input image, but our method still achieves perceptually plausible results consistent with the surrounding image. Note that the motivations of `Pix2PixHD` and our model are different, as `Pix2PixHD` performs image generation from scratch while our focus is local image manipulation.

### 4.3 Qualitative Analysis

**Semantic object manipulation.** To demonstrate how our hierarchical model manipulates structure and style of an object depending on the context, we conduct qualitative analysis in various settings. In Figure 5, we present the manipulation results by moving the same bounding box of a car to different locations in an image. As we see in the figure, our model generates a car with diverse but

reasonably-looking shape and appearance when we move its bounding box from one side of the road to another. Interestingly, the shape, orientation, and appearance of the car also change according to the scene layout and shadow in the surrounding regions. Figure 6 illustrates generation results in more diverse contexts. It shows that our model generates appropriate structure and appearance of the object considering the context (*e.g.* occlusion with other objects, interaction with the scene, etc). In addition to generating object matching the surroundings, we can also easily extend our framework to allow users to directly control object style, which we demonstrate below.

**Extension to style manipulation** Although we focused on addition and deletion of objects as manipulation tasks, we can easily extend our framework to add control over object styles by conditioning the image generator with additional style vector $\mathbf{s}$ by $G^I(\hat{M}, I, \mathbf{s})$. To demonstrate this capability, we define the style vector as a mean color of the object, and synthesized objects while changing the input color (Figure 7). The results show that our model successfully synthesizes various objects with the specified color while retaining other parts of the images unchanged. Modeling more complicated styles (*e.g.* texture) can be achieved by learning a style-encoder $\mathbf{s} = E(X)$ where $X$ is an object template that user can select, but we leave it as a future work.

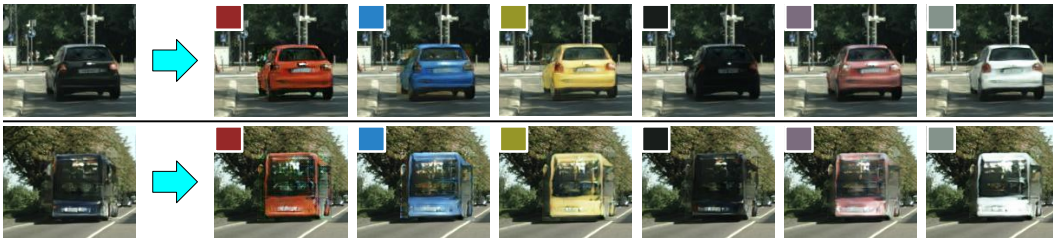

Figure 7: Controlling object color with style vector (left-upper corners indicate conditions).

**Interactive and data-driven image editing.** As one of the key applications, we perform interactive image manipulation by adding, removing, and moving object bounding boxes. Figure 8 illustrates the results. It shows that our method generates reasonable semantic layouts and images that smoothly augment content of the original image. In addition to interactive manipulation, we can also automate the manipulation process by sampling bounding boxes in an image in a data-driven manner. To demonstrate this idea, we present an application of data-driven image manipulation in Figure 9. In this demo, we implement box sampling using a simple non-parametric approach; Given a query image, we first compute its nearest neighbor from the training set based on low-level similarity. Then we compute the geometric transformation between a query and a training image using SIFT Flow [15], and move object bounding boxes from one scene to another based on the scene-level transformation. As shown in the figure, the proposed methods reasonably sample boxes from appropriate locations. However, directly placing the object (or its mask) may lead to unnatural manipulation due to mismatches in scene configuration (*e.g.* occlusion and orientation). Our hierarchical model generates objects adaptive to the new context.

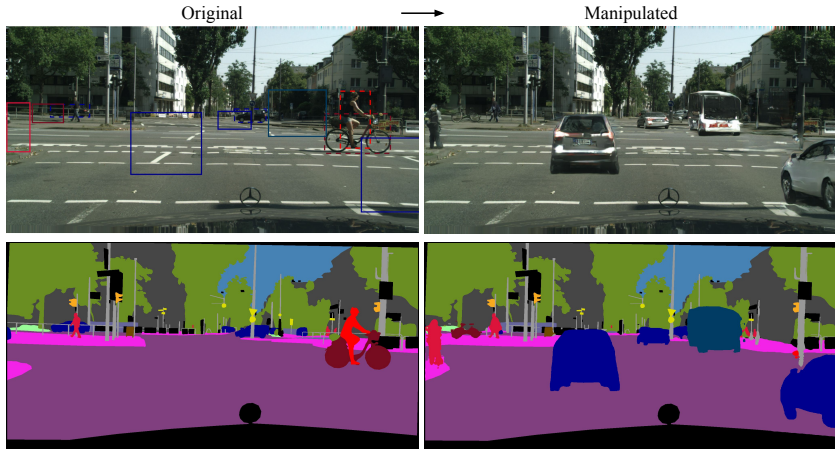

Figure 8: Examples of manipulation of multiple objects in images. The line style indicates manipulation operation (solid: addition, doted: deletion) and the color indicates the object class.

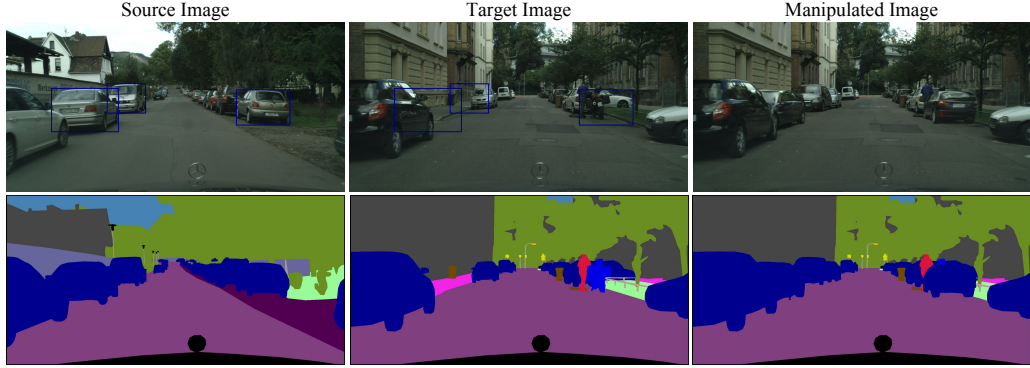

Figure 9: Example of data-driven image manipulation. We manipulate the target image by transferring bounding boxes from source image (blue boxes).

**Results on indoor scene dataset.**    In addition to Cityscape dataset, we conduct qualitative experiments on bedroom images using ADE20K datasets [31]. Figure 10 illustrates the interactive image manipulation results. Since objects in the indoor images involve much more diverse categories and appearances, generating appropriate object shapes and textures aligned with other components in a scene is much more challenging than the street images. We observe that the generated objects by the proposed method usually are looking consistent the surrounding context.

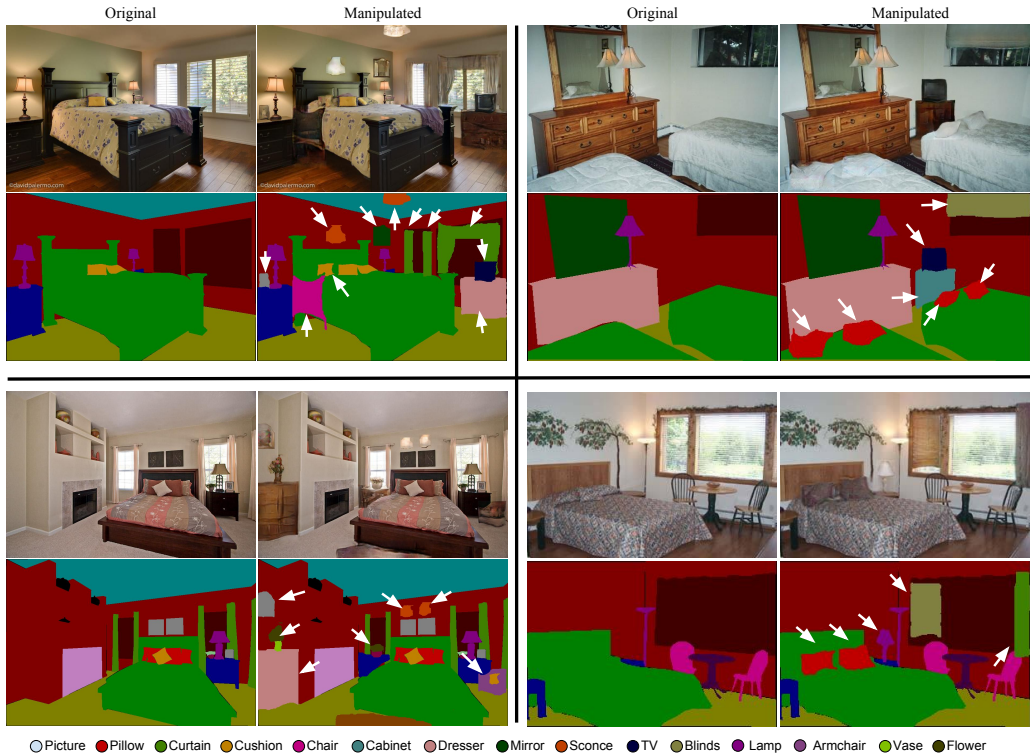

Figure 10: Examples of image manipulation results on indoor images.

## 5   Conclusions

In this paper, we presented a hierarchical framework for semantic image manipulation. We first learn to generate the pixel-wise semantic label maps given the initial object bounding boxes. Then we learn to generate the manipulated image from the predicted label maps. Such framework allows the user to manipulate images at object-level by adding, removing, and moving an object bounding box at a time. Experimental evaluations demonstrate the advantages of the hierarchical manipulation framework over existing image generation and context hole-filing models, both qualitatively and quantitatively. We further demonstrate its practical benefits in semantic object manipulation, interactive image editing and data-driven image editing.

**Acknowledgement**    This work was supported in part by ONR N00014-13-1-0762, NSF CAREER IIS-1453651, DARPA Explainable AI (XAI) program #313498, Sloan Research Fellowship, and Adobe Research Fellowship and Google PhD Fellowship to X. Yan.

## Footnotes

[1]We used $c = 0$ to indicate a deletion operation, where the model fills the labels with surroundings.

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
