[Supplementary Material]

# A Implementation details

**Structure generator** The structure generator takes the semantic label maps as input and produces both foreground object and background context in two separate streams. Overall, the structure generator consists of 3 convolutional downsampling layers, followed by 6 residual convolutional blocks, and finally 3 convolutional upsampling layers. The downsampling convolutional layers have 64, 96, 128 channels with filter size of $7 \times 7$, $4 \times 4$ and $4 \times 4$, respectively. All residual convolutional blocks have 256 channels with the filter size of $4 \times 4$. The upsampling convolutional layers have a symmetric structure as the downsampling convolutional layers.

**Image generator** Our image generator follows the similar architecture used in high-resolution image synthesis [3]. It is composed of 3 convoltional downsampling layers followed by 9 residual blocks and 3 upsampling layers with transposed convolution. All convolutional blocks are implemented by $3 \times 3$ convolutional filters. For discriminator, we used Patch-GAN [1] style network. Similar to [3], we used two discriminators in multiple scales that generate predictions on real or fake over $70 \times 70$ and $140 \times 140$ local image patches, respectively.

# B More quantitative results

**Ablation study using the predicted layout** Table S.1 presents an additional ablation study of our method, which corresponds to the extension of Table 1 in the main paper. In this experiment, we compare the manipulation performance of various image generators using the *predicted* layout. We observe the same trend shown in Table 1, where our method using the two encoder streams (TwoStream) consistently outperforms other variants in all measures.

| | Layout | SSIM | Segmentation (%) | Human eval. (%) |
|---|---|---|---|---|
| SingleStream-Image | - | 0.285 | 59.6 | 20.0 |
| SingleStream-Layout | Predicted | 0.268 | 69.4 | 10.4 |
| SingleStream-Concat | Predicted | 0.296 | 76.3 | 29.9 |
| TwoStream | Predicted | **0.299** | **77.8** | **40.0** |

Table S.1: Comparisons between variants of the proposed method.

**Hierarchical manipulation *vs.* Drag-and-drop** In this paper, we argue that hierarchical manipulation of objects is more appropriate than copy-and-paste of objects in training data, since the manipulation task requires to adapt both shape and appearance of objects based on its surrounding context in a reference image. To clarify this point, we implement drag-and-drop baseline and compare it against ours through human evaluation. Specifically, we sample bounding boxes from testing images, and obtain the nearest neighbor object from a training set using the algorithm presented in the data-driven image editing experiment in Section 4.3. Then we copy and paste the object after applying automatic color adjustment [2]. We compare it against ours using AMT by asking 300 annotators to rank two methods over 100 samples, where our method was favored over the baseline in *76.33%*. As presented in Figure S.1, simple drag-and-drop of the objects sometimes leads to the objects mismatching with the context. For instance, it has to adjust shapes considering the occlusion with other objects (first row) or align its orientation with others (second row). On the other hand, our method generates the appropriate shapes and appearances aligned with other parts of the scene, which leads to plausible manipulation results.

Figure S.1: Examples of ours *vs.* drag-and-drop.

# C More qualitative results

**Qualitative comparison to other methods** Figure S.2 presents qualitative comparison to the existing image manipulation methods presented in Table 2. As it shows, results from both ContextEncoder and ContextEncoder++ are not visually appealing as the generations are usually in a low-quality. Pix2PixHD produces high-quality synthesis results but they are easily distinguishable from the surroundings as the generation is only based on semantic layout. Due to this property, the segmentation performance of Pix2PixHD is higher than ours (Table 2), but human tends to prefer our results over Pix2PixHD because our method produces visually more natural manipulation results.

Figure S.2: Qualitative comparisons to the existing manipulation methods presented in Table 2.

**Semantic object manipulation** Figure S.3 illustrates semantic object manipulation results, which corresponds to Figure 6 of the main paper. As shown in the figure, the proposed method generates structure and style of the objects adaptively depending on the context. For instance, the model generates the shape of the vehicles (car, truck, and bus) in such a way that its orientations are aligned with surrounding geometric environments (*e.g.* sidewalks and road). Beyond geometric constraints, the model also learns biases in the scene and generates shapes based on it (*e.g.* a person *standing* on a sidewalk *vs.* a person *walking* on a crosswalk). Also, when there are objects overlapped with the user-defined bounding box, the model internally figures out the order and generates shapes considering the occlusion between objects (*e.g.* car behind person and pole).

Figure S.3: Generation results in diverse context.

**Interactive and data-driven image manipulation** We present more examples on interactive image editing, which correspond to Figure 8 in the main paper. Figure S.4 and S.5 illustrate the editing results obtained by iteratively adding, moving and removing the object bounding boxes. For better understanding, we also visualize the manipulation results at each step. More examples on interactive image editing are illustrated in Figure S.6. We can see that the proposed method generates reasonable manipulation in various scenes.

Figure S.7 illustrates more examples on data-driven image editing, which correspond to Figure 9 of the main paper.

Figure S.4: Interactive image manipulation. Given an input image, the user performs interactive image manipulation by adding or removing one object at a time. The top part demonstrates the object-level manipulation while the bottom part shows the original input image and the manipulation result in the end. The bounding box style indicates manipulation operation (solid: addition, doted: deletion) and the color of bounding box indicates the object class.

Figure S.5: Interactive image manipulation. Given an input image, the user performs interactive image manipulation by adding or removing one object at a time. The top part demonstrates the object-level manipulation while the bottom part shows the original input image and the manipulation result in the end. The bounding box style indicates manipulation operation (solid: addition, doted: deletion) and the color of bounding box indicates the object class.

Original → Manipulated

Figure S.6: More examples on interactive image editing.

Figure S.7: Example of data-driven image manipulation. We manipulate the target image by transferring bounding boxes from source image (blue boxes).

**Results on indoor scene dataset.** Figure S.8 illustrates more interactive image editing results on ADE20K bedroom dataset, which corresponds to Figure 10 of the main paper. Compared to Cityscape dataset, the indoor images are composed of much more densly populated objects from various categories exhibiting diverse appearances. Despite of such challenges, the proposed method generates visually plausible objects aligned with surrounding context in many examples (*e.g.* alignement of the manipulated objects with orientation of room layout or other furniture).

Figure S.8: Example of interactive image editing on ADE20K indoor scene datset [4]. Manipulated parts of the images are indicated by arrows.