[Reviews · NeurIPS 2018]

Reviewer 1



In this paper a new method for image manipulation is proposed. The proposed method incorporates a hierarchical framework and provides both interactive and automatic semantic object-level image manipulation. In the interactive manipulation setting, the user can select a bounding box where image editing for adding and removing objects will be applied. The proposed network architecture consists of a foreground output stream which produces the predictions on binary object mask and a background output stream for producing per-pixel label maps. As the result, the proposed image manipulation method generates output image by filling in the pixel-level textures guided by the semantic layout. The manipulation operations are only ‘insertion’ and ‘deletion’ and other operations such as change of color and style are not supported. The paper is well-written and clear; Figures and qualitative examples convey the idea. As to investigate the effectiveness of the single-stream vs two-stream using ground truth or predicted layout ablation study is conducted. The performance of the proposed method is compared with prior works of [21] and [32] and human study is conducted via AMT. Based on the results summarized in Table 2, the proposed method has higher SSIM and obtains higher human evaluation compared to the prior works and baseline. However, as for the the segmentation accuracy the prior work of [32] obtains higher accuracy. While the proposed method is interesting, the evaluation is only conducted on outdoor street images and only two selected examples are shown in the indoor scenes in supplemental. In light of the high quality performance of the previous works on image generation such as [4] and [41], image generation in the context of indoor scenes is more subject to evaluation as such indoor scenes involve more visual diversity and contain larger set of object categories for manipulation. Most of the examples shown in the paper demonstrate the quality of adding or removing cars and person in street scenes which have high biases in terms of pose and shape. The question that remains is that how much object diversity (in terms of both category and appearance differences) such image manipulation technique can handle.

Reviewer 2



+ task and proposed model seems novel + good evaluation - evaluation on own task - task not sufficiently motivated The paper is well written and easy to follow. The task of image manipulation through bounding boxes seems somewhat novel, and the proposed model going from a box input through an intermediate semantic segmentation is new (although semantics has been used before to generate images). The evaluation is extensive and contains several human studies. Unfortunately, the comparison is not quite fair, as none of the baselines know the identity of the object removed (or the object to be generated). Have the authors considered training class specific baselines? Finally, the authors didn't fully convince me that drawing labeled bounding boxes is a natural interface for image manipulation. In fact, there are many alternatives that seem more natural at first glance (e.g. drag and drop of existing dataset objective + color adjustment, direct color manipulation, ...). It would help if the authors could provide some additional motivation why this interface is particularly compelling (ideally with some experimental results). Is the interface easier to interact with? Does it give the user more freedom? ... Minor: * Have the authors considered merging the two streams in fig 2? What does the object stream offer, over a context stream that also produces the foreground mask? * Is the object stream conditioned on the class c? There does this conditioning happen?

Reviewer 3



Quality: This paper shows a reasonable technical framework. The main idea of this approach is well supported by benchmarks and some comparisons with other methods. Both tests and results are conclusive in the experiments. Extra information and an interactive demo are also presented for visualization results. Clarity: Overall, the document is properly written, however it has some inconclusive paragraphs. Such as in section 2 - “Related Work”, where three groups of previous works are introduced, but the authors do not show conclusive ideas about the direct link between the related methods and the proposed one. Furthermore, cross-references in some figures can be improved with a more explicit reference. (line 221: “As shown in the table ...”; line 257: “In this demo ...”.) Originality: The proposed method is a novel combination of well-known techniques. Some differences are presented between the proposed approach and some similar methods, Pix2PixHD and Context Encoder. Significance: The presented method shows plausible results in both extra provided information and an interactive demo, nevertheless the current results show a slight improvement.